# Scalable Bayesian Learning of Recurrent Neural Networks for Language Modeling

## Abstract

Recurrent neural networks (RNNs) have shown promising performance for language modeling. However, traditional training of RNNs using back-propagation through time often suffers from overfitting. One reason for this is that stochastic optimization (used for large training sets) does not provide good estimates of model uncertainty. This paper leverages recent advances in stochastic gradient Markov Chain Monte Carlo (also appropriate for large training sets) to learn weight uncertainty in RNNs. It yields a principled Bayesian learning algorithm, adding gradient noise during training (enhancing exploration of the model-parameter space) and model averaging when testing. Extensive experiments on various RNN models and across a broad range of applications demonstrate the superiority of the proposed approach relative to stochastic optimization.

## 1 Introduction

Language modeling is a fundamental task, used for example to predict the next word or character in a text sequence given the context. Recently, recurrent neural networks (RNNs) have shown promising performance on this task (Mikolov et al., 2010; Sutskever et al., 2011). RNNs with Long Short-Term Memory (LSTM) units (Hochreiter and Schmidhuber, 1997) have emerged as a popular architecture, due to their representational power and effectiveness at capturing long-term dependencies.

RNNs are usually trained via back-propagation through time (Werbos, 1990), using stochastic optimization methods such as stochastic gradient descent (SGD) (Robbins and Monro, 1951); stochastic methods of this type are particularly important for training with large data sets. However, this approach often provides a *maximum a posteriori* (MAP) estimate of model parameters. The MAP solution is a single point estimate, ignoring weight uncertainty (Blundell et al., 2015; Hernández-Lobato and Adams, 2015). Natural language often exhibits significant variability, and hence such a point estimate may make over-confident predictions on test data.

To alleviate overfitting RNNs, good regularization is known as a key factor to successful applications. In the neural network literature, Bayesian learning has been proposed as a principled method to impose regularization and incorporate model uncertainty (MacKay, 1992; Neal, 1995), by imposing prior distributions on model parameters. Due to the intractability of posterior distributions in neural networks, Hamiltonian Monte Carlo (HMC) (Neal, 1995) has been used to provide sample-based approximations to the true posterior. Despite the elegant theoretical property of asymptotic convergence to the true posterior, HMC and other conventional Markov Chain Monte Carlo methods are not scalable to large training sets.

This paper seeks to scale up Bayesian learning of RNNs to meet the challenge of the increasing amount of "big" sequential data in natural language processing, leveraging recent advances in *stochastic* gradient Markov Chain Monte Carlo (SG-MCMC) algorithms (Welling and Teh, 2011; Chen et al., 2014; Ding et al., 2014; Li et al., 2016a). Specifically, instead of training a single network, SG-MCMC is employed to train an *ensemble* of networks, where each network has its parameters drawn from a shared posterior distribution. This is implemented by adding additional gradient noise during training and utilizing model

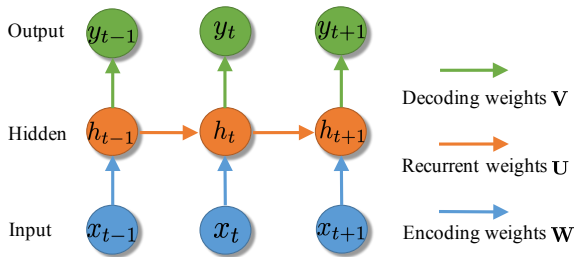

Figure 1: Illustration of different weight learning strategies in a single-hidden-layer RNN. Stochastic optimization used for MAP estimation puts fixed values on all weights. Naive dropout is allowed to put weight uncertainty only on encoding and decoding weights, and fixed values on recurrent weights. The proposed SG-MCMC scheme imposes distributions on all weights.

averaging when testing.

This simple procedure has the following salutary properties for training neural networks: (*i*) When training, the injected noise encourages model-parameter trajectories to better explore the parameter space. This procedure was also empirically found effective in (Neelakantan et al., 2016). (*ii*) Model averaging when testing alleviates overfitting and hence improves generalization, transferring uncertainty in the learned model parameters to subsequent prediction. (*iii*) In theory, both asymptotic and non-asymptotic consistency properties of SG-MCMC methods in posterior estimation have been recently established to guarantee convergence (Chen et al., 2015a; Teh et al., 2016). (*iv*) SG-MCMC is scalable; it shares the same level of computational cost as SGD in training, by only requiring the evaluation of gradients on a small mini-batch. To the authors' knowledge, RNN training using SG-MCMC has not been investigated previously, and is a contribution of this paper. We also perform extensive experiments on several natural language processing tasks, demonstrating the effectiveness of SG-MCMC for RNNs, including character/word-level language modeling, image captioning and sentence classification.

## 2 Related Work

Several scalable Bayesian learning methods have been proposed recently for neural networks. These come in two broad categories: stochastic variational inference (Graves, 2011; Blundell et al., 2015; Hernández-Lobato and Adams, 2015) and SG-MCMC methods (Korattikara et al., 2015; Li et al., 2016a). While prior work focuses on feed-forward neural networks, there has been little if any research reported for RNNs using SG-MCMC.

Dropout (Hinton et al., 2012; Srivastava et al., 2014) is a commonly used regularization method for training neural networks. Recently, several works have studied how to apply dropout to RNNs (Pachitariu and Sahani, 2013; Bayer et al., 2013; Pham et al., 2014; Zaremba et al., 2014; Bluche et al., 2015; Moon et al., 2015; Semeniuta et al., 2016; Gal and Ghahramani, 2016b). Among them, naive dropout (Zaremba et al., 2014) can impose weight uncertainty only on *encoding weights* (those that connect input to hidden units) and *decoding weights* (those that connect hidden units to output), but not the *recurrent weights* (those that connect consecutive hidden states). It has been concluded that noise added in the recurrent connections leads to model instabilities, hence disrupting the RNN's ability to model sequences.

Dropout has been recently shown to be a variational approximation technique in Bayesian learning (Gal and Ghahramani, 2016a; Kingma et al., 2015). Based on this, (Gal and Ghahramani, 2016b) proposed a new variant of dropout that can be successfully applied to recurrent layers, where the same dropout masks are shared along time for encoding, decoding and recurrent weights, respectively. Alternatively, we focus on SG-MCMC, which can be viewed as the Bayesian interpretation of dropout from the perspective of posterior sampling (Li et al., 2016b); this also allows imposition of model uncertainty on recurrent layers, enhancing performance. A comparison of naive dropout and SG-MCMC is illustrated in Fig. 1.

## 3 Recurrent Neural Networks

### 3.1 RNN as Bayesian Predictive Models

Consider data $\mathcal{D} = \{\mathbf{D}_1, \cdots, \mathbf{D}_N\}$, where $\mathbf{D}_n \triangleq (\mathbf{X}_n, \mathbf{Y}_n)$, with input $\mathbf{X}_n$ and output $\mathbf{Y}_n$. Our goal is to learn model parameters $\boldsymbol{\theta}$ to best characterize the relationship from $\mathbf{X}_n$ to $\mathbf{Y}_n$, with corresponding data likelihood $p(\mathcal{D}|\boldsymbol{\theta}) = \prod_{n=1}^{N} p(\mathbf{D}_n|\boldsymbol{\theta})$. In Bayesian statistics, one sets a prior on $\boldsymbol{\theta}$ via distribution $p(\boldsymbol{\theta})$. The posterior $p(\boldsymbol{\theta}|\mathcal{D}) \propto p(\boldsymbol{\theta})p(\mathcal{D}|\boldsymbol{\theta})$ reflects the belief concerning the model parameter distribution after observing the data. Given a test input $\tilde{\mathbf{X}}$ (with missing output $\tilde{\mathbf{Y}}$), the uncertainty learned in training is transferred to prediction, yielding the posterior

predictive distribution:

$$p(\tilde{\mathbf{Y}}|\tilde{\mathbf{X}}, \mathcal{D}) = \int_{\boldsymbol{\theta}} p(\tilde{\mathbf{Y}}|\tilde{\mathbf{X}}, \boldsymbol{\theta}) p(\boldsymbol{\theta}|\mathcal{D}) \mathrm{d}\boldsymbol{\theta} \,. \quad (1)$$

When the input is a sequence, RNNs may be used to parameterize the input-output relationship. Specifically, consider input sequence $\mathbf{X} = \{\boldsymbol{x}_1, \ldots, \boldsymbol{x}_T\}$, where $\boldsymbol{x}_t$ is the input data vector at time $t$. There is a corresponding hidden state vector $\boldsymbol{h}_t$ at each time $t$, obtained by recursively applying the *transition function* $\boldsymbol{h}_t = \mathcal{H}(\boldsymbol{h}_{t-1}, \boldsymbol{x}_t)$ (specified in Section 3.2; see Fig. 1). The ouput $\mathbf{Y}$ differs depending on the application: a sequence $\{\boldsymbol{y}_1, \ldots, \boldsymbol{y}_T\}$ in language modeling or a discrete label in sentence classification. In RNNs the corresponding *decoding function* is $p(\boldsymbol{y}|\boldsymbol{h})$, described in Section 3.3.

## 3.2 RNN Architectures

The transition function $\mathcal{H}(\cdot)$ can be implemented with a *gated* activation function, such as Long Short-Term Memory (LSTM) (Hochreiter and Schmidhuber, 1997) or a Gated Recurrent Unit (GRU) (Cho et al., 2014). Both the LSTM and GRU have been proposed to address the issue of learning long-term sequential dependencies.

**Long Short-Term Memory**   The LSTM architecture addresses the problem of learning long-term dependencies by introducing a *memory cell*, that is able to preserve the state over long periods of time. Specifically, each LSTM unit has a cell containing a state $\boldsymbol{c}_t$ at time $t$. This cell can be viewed as a memory unit. Reading or writing the cell is controlled through sigmoid gates: input gate $\boldsymbol{i}_t$, forget gate $\boldsymbol{f}_t$, and output gate $\boldsymbol{o}_t$. The hidden units $\boldsymbol{h}_t$ are updated as

$$\boldsymbol{i}_t = \sigma(\mathbf{W}_i \boldsymbol{x}_t + \mathbf{U}_i \boldsymbol{h}_{t-1} + \boldsymbol{b}_i)\,,$$
$$\boldsymbol{f}_t = \sigma(\mathbf{W}_f \boldsymbol{x}_t + \mathbf{U}_f \boldsymbol{h}_{t-1} + \boldsymbol{b}_f)\,,$$
$$\boldsymbol{o}_t = \sigma(\mathbf{W}_o \boldsymbol{x}_t + \mathbf{U}_o \boldsymbol{h}_{t-1} + \boldsymbol{b}_o)\,,$$
$$\tilde{\boldsymbol{c}}_t = \tanh(\mathbf{W}_c \boldsymbol{x}_t + \mathbf{U}_c \boldsymbol{h}_{t-1} + \boldsymbol{b}_c)\,,$$
$$\boldsymbol{c}_t = \boldsymbol{f}_t \odot \boldsymbol{c}_{t-1} + \boldsymbol{i}_t \odot \tilde{\boldsymbol{c}}_t\,,$$
$$\boldsymbol{h}_t = \boldsymbol{o}_t \odot \tanh(\boldsymbol{c}_t)\,,$$

where $\sigma(\cdot)$ denotes the logistic sigmoid function, and $\odot$ represents the element-wise matrix multiplication operator. $\mathbf{W}_{\{i,f,o,c\}}$ are *encoding weights*, and $\mathbf{U}_{\{i,f,o,c\}}$ are *recurrent weights*, as shown in Fig. 1. $\boldsymbol{b}_{\{i,f,o,c\}}$ are bias terms.

**Variants**   Similar to the LSTM unit, the GRU also has gating units that modulate the flow of information inside the hidden unit. It has been shown that a GRU can achieve similar performance to an LSTM in sequence modeling (Chung et al., 2014). We specify the GRU in the Supplementary Material.

The LSTM can be extended to the bidirectional LSTM and multilayer LSTM. A bidirectional LSTM consists of two LSTMs that are run in parallel: one on the input sequence and the other on the reverse of the input sequence. At each time step, the hidden state of the bidirectional LSTM is the concatenation of the forward and backward hidden states. In multilayer LSTMs, the hidden state of an LSTM unit in layer $\ell$ is used as input to the LSTM unit in layer $\ell+1$ at the same time step.

## 3.3 Applications

The proposed Bayesian framework can be applied to any RNN model; we focus on the following tasks to demonstrate the ideas.

**Language Modeling**   In word-level language modeling, the input to the network is a sequence of words, and the network is trained to predict the next word in the sequence with a softmax classifier. Specifically, for a length-$T$ sequence, denote $\boldsymbol{y}_t = \boldsymbol{x}_{t+1}$ for $t = 1, \ldots, T-1$. $\boldsymbol{x}_1$ and $\boldsymbol{y}_T$ are always set to a special START and END token, respectively. At each time $t$, there is a decoding function $p(\boldsymbol{y}_t|\boldsymbol{h}_t) = \text{softmax}(\mathbf{V}\boldsymbol{h}_t)$ to compute the distribution over words, where $\mathbf{V}$ are the *decoding weights* (the number of rows of $\mathbf{V}$ corresponds to the number of words/characters). We also extend this basic language model to consider other applications: (*i*) a *character-level language model* can be specified in a similar manner by replacing words with characters (Karpathy et al., 2016). (*ii*) *Image captioning* can be considered as a conditional language modeling problem, in which we learn a generative language model of the caption conditioned on an image (Vinyals et al., 2015).

**Sentence Classification**   Sentence classification aims to assign a semantic category label $\boldsymbol{y}$ to a whole sentence $\mathbf{X}$. This is usually implemented through applying the decoding function once at the end of sequence: $p(\boldsymbol{y}|\boldsymbol{h}_T) = \text{softmax}(\mathbf{V}\boldsymbol{h}_T)$, where the final hidden state of a RNN $\boldsymbol{h}_T$ is often considered as the summary of the sentence (here

the number of rows of $\mathbf{V}$ corresponds to the number of classes).

## 4 Scalable Learning with SG-MCMC

### 4.1 The Pitfall of Stochastic Optimization

Typically there is no closed-form solution for the posterior $p(\boldsymbol{\theta}|\mathcal{D})$, and traditional Markov Chain Monte Carlo (MCMC) methods (Neal, 1995) scale poorly for large $N$. To ease the computational burden, stochastic optimization is often employed to find the MAP solution. This is equivalent to minimizing an objective of regularized loss function $U(\boldsymbol{\theta})$ that corresponds to a (non-convex) model of interest: $\boldsymbol{\theta}_{\text{MAP}} = \arg\min U(\boldsymbol{\theta})$, $U(\boldsymbol{\theta}) = -\log p(\boldsymbol{\theta}|\mathcal{D})$. The expectation in (1) is approximated as:

$$p(\tilde{\mathbf{Y}}|\tilde{\mathbf{X}}, \mathcal{D}) = p(\tilde{\mathbf{Y}}|\tilde{\mathbf{X}}, \boldsymbol{\theta}_{\text{MAP}}) . \qquad (2)$$

Though simple and effective, this procedure largely loses the benefit of the Bayesian approach, because the uncertainty on weights is ignored. To more accurately approximate (1), we employ stochastic gradient (SG) MCMC (Welling and Teh, 2011).

### 4.2 Large-scale Bayesian Learning

The negative log-posterior is

$$U(\boldsymbol{\theta}) \triangleq -\log p(\boldsymbol{\theta}) - \sum_{n=1}^{N} \log p(\mathbf{D}_n|\boldsymbol{\theta}). \qquad (3)$$

In optimization, $E = -\sum_{n=1}^{N} \log p(\mathbf{D}_n|\boldsymbol{\theta})$ is typically referred to as the loss function, and $R \propto -\log p(\boldsymbol{\theta})$ as a regularizer.

For large $N$, stochastic approximations are often employed:

$$\tilde{U}_t(\boldsymbol{\theta}) \triangleq -\log p(\boldsymbol{\theta}) - \frac{N}{M} \sum_{m=1}^{M} \log p(\mathbf{D}_{i_m}|\boldsymbol{\theta}), \qquad (4)$$

where $\mathcal{S}_m = \{i_1, \cdots, i_M\}$ is a *random* subset of the set $\{1, 2, \cdots, N\}$, with $M \ll N$. The gradient on this mini-batch is denoted as $\tilde{\boldsymbol{f}}_t = \nabla \tilde{U}_t(\boldsymbol{\theta})$, which is an unbiased estimate of the true gradient. The evaluation of (4) is cheap even when $N$ is large, allowing one to efficiently collect a sufficient number of samples in large-scale Bayesian learning, $\{\boldsymbol{\theta}_s\}_{s=1}^{S}$, where $S$ is the number of samples (this will be specified later). These samples are used to construct a sample-based estimation to the expectation in (1):

Table 1: SG-MCMC algorithms and their optimization counterparts. Algorithms in the same row share similar characteristics.

| Algorithms | SG-MCMC | Optimization |
|---|---|---|
| *Basic* | SGLD | SGD |
| *Precondition* | pSGLD | RMSprop/Adagrad |
| *Momentum* | SGHMC | momentum SGD |
| *Thermostat* | SGNHT | Santa |

$$p(\tilde{\mathbf{Y}}|\tilde{\mathbf{X}}, \mathcal{D}) \approx \frac{1}{S} \sum_{s=1}^{S} p(\tilde{\mathbf{Y}}|\tilde{\mathbf{X}}, \boldsymbol{\theta}_s) . \qquad (5)$$

The finite-time estimation errors of SG-MCMC methods are bounded (Chen et al., 2015a), which guarantees (5) is an unbiased estimate of (1) asymptotically under appropriate decreasing step-sizes.

### 4.3 SG-MCMC Algorithms

SG-MCMC and stochastic optimization are parallel lines of work, designed for different purposes; their relationship has recently been revealed in the context of deep learning. The most basic SG-MCMC algorithm has been applied to Langevin dynamics, and is termed SGLD (Welling and Teh, 2011). To help convergence, a momentum term has been introduced in SGHMC (Chen et al., 2014), a "thermostat" has been devised in SGNHT (Ding et al., 2014) and preconditioners have been employed in pSGLD (Li et al., 2016a). These SG-MCMC algorithms often share similar characteristics with their counterpart approaches from the optimization literature such as the momentum SGD, Santa (Chen et al., 2016) and RMSprop/Adagrad (Tieleman and Hinton, 2012; Duchi et al., 2011). The interrelationships between SG-MCMC and optimization-based approaches are summarized in Table 1.

**SGLD** Stochastic Gradient Langevin Dynamics (SGLD) (Welling and Teh, 2011) draws posterior samples, with updates

$$\boldsymbol{\theta}_t = \boldsymbol{\theta}_{t-1} - \eta_t \tilde{\boldsymbol{f}}_{t-1} + \sqrt{2\eta_t} \boldsymbol{\xi}_t , \qquad (6)$$

where $\eta_t$ is the learning rate, and $\boldsymbol{\xi}_t \sim \mathcal{N}(\mathbf{0}, \mathbf{I}_p)$ is a standard Gaussian random vector. SGLD is the SG-MCMC analog to stochastic gradient descent (SGD), whose parameter updates are given by:

$$\boldsymbol{\theta}_t = \boldsymbol{\theta}_{t-1} - \eta_t \tilde{\boldsymbol{f}}_{t-1} . \qquad (7)$$

SGD is guaranteed to converge to a local minimum under mild conditions (Bottou, 2010). The

---

**Algorithm 1:** pSGLD

**Input**: Default hyperparameter settings:
$\eta_t = 1 \times 10^{-3}, \lambda = 10^{-8}, \beta_1 = 0.99.$

**Initialize**: $\boldsymbol{v}_0 \leftarrow \boldsymbol{0}, \boldsymbol{\theta}_1 \sim \mathcal{N}(0, \mathbf{I})$ ;

**for** $t = 1, 2, \ldots, T$ **do**

$\quad$ % Estimate gradient from minibatch $\mathcal{S}_{\mathrm{t}}$

$\quad \tilde{\boldsymbol{f}}_t = \nabla \tilde{U}_t(\boldsymbol{\theta})$;

$\quad$ % Preconditioning

$\quad \boldsymbol{v}_t \leftarrow \beta_1 \boldsymbol{v}_{t-1} + (1 - \beta_1) \tilde{\boldsymbol{f}}_t \odot \tilde{\boldsymbol{f}}_t$;

$\quad \mathbf{G}_t^{-1} \leftarrow \mathrm{diag}\left( \mathbf{1} \oslash \left( \lambda \mathbf{1} + \boldsymbol{v}_t^{\frac{1}{2}} \right) \right)$;

$\quad$ % Parameter update

$\quad \boldsymbol{\xi}_t \sim \mathcal{N}(0, \eta_t \mathbf{G}_t^{-1})$;

$\quad \boldsymbol{\theta}_{t+1} \leftarrow \boldsymbol{\theta}_t + \frac{\eta_t}{2} \mathbf{G}_t^{-1} \tilde{\boldsymbol{f}}_t + \boldsymbol{\xi}_t$;

**end**

---

additional Gaussian term in SGLD helps the learning trajectory to explore the parameter space to approximate posterior samples, instead of obtaining a local minimum.

**pSGLD** Preconditioned SGLD (pSGLD) (Li et al., 2016a) was proposed recently to improve the mixing of SGLD. It utilizes magnitudes of recent gradients to construct a diagonal preconditioner to approximate the Fisher information matrix, and thus adjusts to the local geometry of parameter space by equalizing the gradients so that a constant stepsize is adequate for all dimensions. This is important for RNNs, whose parameter space often exhibits *pathological curvature* and *saddle points* (Pascanu et al., 2013), resulting in slow mixing. There are multiple choices of preconditioners; similar ideas in optimization include Adagrad (Duchi et al., 2011), Adam (Kingma and Ba, 2015) and RMSprop (Tieleman and Hinton, 2012). An efficient version of pSGLD, adopting RMSprop as the preconditioner $\mathbf{G}$, is summarized in Algorithm 1, where $\oslash$ denotes elementwise matrix division. When the preconditioner is fixed as the identity matrix, the method reduces to SGLD.

### 4.4 Understanding SG-MCMC

To further understand SG-MCMC, we show its close connection to dropout/dropConnect (Srivastava et al., 2014; Wan et al., 2013). These methods improve the generalization ability of deep models, by randomly adding binary/Gaussian noise to the local units or global weights. For neural networks with the nonlinear function $q(\cdot)$ and consecutive layers $\boldsymbol{h}_1$ and $\boldsymbol{h}_2$, dropout and dropConnect are denoted as:

$$\text{Dropout:} \qquad \boldsymbol{h}_2 = \boldsymbol{\xi}_0 \odot q(\boldsymbol{\theta} \boldsymbol{h}_1),$$
$$\text{DropConnect:} \qquad \boldsymbol{h}_2 = q((\boldsymbol{\xi}_0 \odot \boldsymbol{\theta}) \boldsymbol{h}_1),$$

where the injected noise $\boldsymbol{\xi}_0$ can be binary-valued with dropping rate $p$ or its equivalent Gaussian form (Wang and Manning, 2013):

$$\text{Binary noise:} \qquad \boldsymbol{\xi}_0 \sim \mathrm{Ber}(p),$$
$$\text{Gaussian noise:} \qquad \boldsymbol{\xi}_0 \sim \mathcal{N}(1, \frac{p}{1-p}).$$

Note that $\boldsymbol{\xi}_0$ is defined as a vector for dropout, and a matrix for dropConnect. By combining dropConnect and Gaussian noise from the above, we have the update rule (Li et al., 2016b):

$$\boldsymbol{\theta}_{t+1} = \boldsymbol{\xi}_0 \odot \boldsymbol{\theta}_t - \frac{\eta}{2} \tilde{\boldsymbol{f}}_t = \boldsymbol{\theta}_t - \frac{\eta}{2} \tilde{\boldsymbol{f}}_t + \boldsymbol{\xi}_0', \quad (8)$$

where $\boldsymbol{\xi}_0' \sim \mathcal{N}\left(0, \frac{p}{(1-p)} \mathrm{diag}(\boldsymbol{\theta}_t^2)\right)$; (8) shows that dropout/ dropConnect and SGLD in (6) share the same form of update rule, with the distinction being that the level of injected noise is different. In practice, the noise injected by SGLD may not be enough. A better way that we find to improve the performance is to jointly apply SGLD and dropout. This method can be interpreted as using SGLD to sample the posterior distribution of a mixture of RNNs, with mixture probability controlled by the dropout rate.

## 5 Experiments

We present results on several tasks, including character/ word-level language modeling, image captioning, and sentence classification. The hyperparameter settings, the initialization of model parameters and model specifications on each dataset are provided in the Supplementary Material.

We do not perform any dataset-specific tuning other than early stopping on validation sets. When dropout is utilized, the dropout rate is set to 0.5. All experiments are implemented in Theano (Theano Development Team, 2016), using a NVIDIA GeForce GTX TITAN X GPU with 12GB memory.

### 5.1 Language Modeling

We first test character-level and word-level language modeling. The setup for each task is as follows.

- Following (Karpathy et al., 2016), we test character-level language modeling on the *War and Peace* (WP) novel. The training/validation/test sets contain 260/32/33 batches, in which there are 100 characters. The vocabulary size is 87, and we consider a 2-hidden-layer RNN of dimension 128.

- The *Penn Treebank* (PTB) corpus (Marcus et al., 1993) is used for word-level language modeling. The dataset adopts the standard split (929K training words, 73K validation words, and 82K test words) and has a vocabulary of size 10K. We train LSTMs of three sizes; these are denoted the small/medium/large LSTM. All LSTMs have two layers and are unrolled for 20 steps. The small, medium and large LSTM has 200, 650 and 1500 units per layer, respectively.

We study the effects of different types of architecture (LSTM/GRU/Vanilla RNN (Karpathy et al., 2016)) on the WP dataset, and effects of different learning algorithms on the PTB dataset. The comparison of test cross-entropy loss on WP is shown in Table 2. We observe that pSGLD consistently outperforms RMSprop. Table 3 summarizes the test set performance on PTB[1]. It is clear that our sampling-based method consistently outperforms the optimization counterpart, where the performance gain mainly comes from adding gradient noise and model averaging. When compared with dropout, SGLD performs better on the small LSTM model, but slightly worse on the medium and large LSTM model. This may imply that dropout is suitable to regularizing large networks, while SGLD exhibits better regularization ability on small networks, partially due to the fact that dropout may inject a higher level of noise during training than SGLD. In order to inject a higher level of noise into SGLD, we empirically apply SGLD and dropout jointly, and found that this provided the best performace on the medium and large LSTM model.

We study three strategies to do model averaging, *i.e.*, *forward collection*, *backward collection* and *thinned collection*. Given samples $(\boldsymbol{\theta}_1, \cdots, \boldsymbol{\theta}_K)$ and the number of samples $S$ used for averaging, *forward collection* refers to using $(\boldsymbol{\theta}_1, \cdots, \boldsymbol{\theta}_S)$ for

---

[1] The results reported here do not match (Zaremba et al., 2014) due to the differences of experimental setup. However, we provide a fair comparison to all methods.

Table 2: Test cross-entropy loss on WP dataset.

| Methods | LSTM | GRU | RNN |
|---------|--------|--------|--------|
| RMSprop | 1.3607 | 1.2759 | 1.4239 |
| pSGLD | **1.3375** | **1.2561** | **1.4093** |

Table 3: Test perplexity on Penn Treebank.

| Methods | Small | Medium | Large |
|---------|-------|--------|-------|
| SGD | 123.85 | 126.31 | 130.25 |
| SGD+Dropout | 136.39 | 100.12 | 97.65 |
| SGLD | **117.36** | 109.14 | 105.86 |
| SGLD+Dropout | 139.54 | **99.58** | **94.03** |

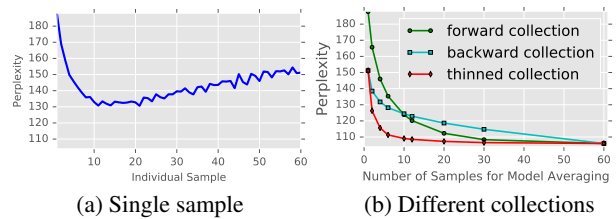

(a) Single sample (b) Different collections

Figure 2: Effects of collected samples.

the evaluation of a test function, *backward collection* refers to using $(\boldsymbol{\theta}_{K-S+1}, \cdots, \boldsymbol{\theta}_K)$, while *thinned collection* chooses samples from $\boldsymbol{\theta}_1$ to $\boldsymbol{\theta}_K$ with interval $K/S$. Fig. 2 plots the effects of these strategies, where Fig. 2(a) plots the perplexity of every single sample, Fig. 2(b) plots the perplexities using the three schemes. Only after 20 samples is a converged perplexity achieved in the thinned collection, while it requires 30 samples for forward collection or 60 samples for backward collection. This is unsurprising, because thinned collection provides a better way to select samples. Nevertheless, averaging of samples provides significantly lower perplexity than using single samples. Note that the overfitting problem in Fig. 2(a) is also alleviated by model averaging.

To better illustrate the benefit of model averaging, we visualize in Fig. 3 the probabilities of each word in a randomly chosen test sentence. The first 3 rows are the results predicted by 3 distinctive model samples, respectively; the bottom row is the result after averaging. Their corresponding perplexities for the test sentence are also shown on the right of each row. The 3 individual samples provide reasonable probabilities. For example, the consecutive words "New York", "stock exchange" and "did not" are assigned with a higher probability. After averaging, we can see a much lower perplexity, as the samples can complement each other. For example, though the second sample can

Table 4: Performance on Flickr8k & Flickr30k in terms of BLEU-1,2,3,4, METEOR, CIDEr, ROUGE-L and perplexity.

| Methods | B-1 | B-2 | B-3 | B-4 | METEOR | CIDEr | ROUGE-L | Perp. |
|---|---|---|---|---|---|---|---|---|
| *Results on Flickr8k* | | | | | | | | |
| RMSprop | 0.640 | 0.427 | 0.288 | 0.197 | 0.205 | 0.476 | 0.500 | 16.64 |
| RMSprop + Dropout | 0.647 | 0.444 | 0.305 | 0.209 | 0.208 | 0.514 | 0.510 | 15.72 |
| pSGLD | **0.669** | **0.463** | **0.321** | **0.224** | **0.214** | **0.535** | **0.522** | 14.29 |
| pSGLD + Dropout | 0.656 | 0.450 | 0.309 | 0.211 | 0.209 | 0.512 | 0.512 | **14.26** |
| *Results on Flickr30k* | | | | | | | | |
| RMSprop | 0.644 | 0.422 | 0.279 | 0.184 | 0.180 | 0.372 | 0.476 | 17.80 |
| RMSprop + Dropout | 0.656 | 0.435 | 0.295 | 0.200 | 0.185 | 0.396 | 0.481 | 18.05 |
| pSGLD | 0.657 | 0.438 | 0.300 | 0.206 | **0.192** | **0.421** | **0.490** | **15.61** |
| pSGLD + Dropout | **0.666** | **0.448** | **0.308** | **0.209** | 0.189 | 0.419 | 0.487 | 17.05 |

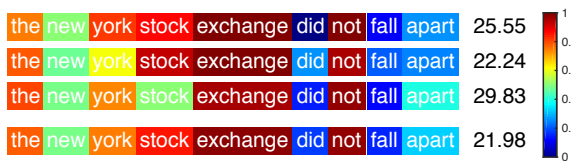

Figure 3: Predictive probabilities obtained by 3 samples and their average. Colors indicate normalized probability of each word. Best viewed in color.

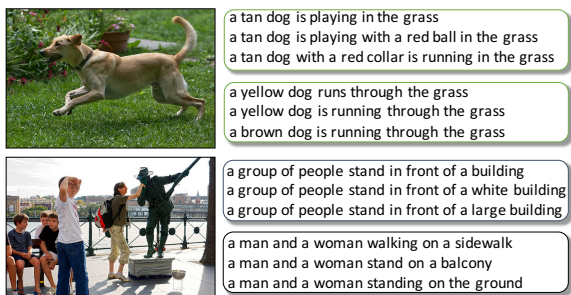

Figure 4: Image captioning with different samples. Left are the given images, right are the corresponding captions. The captions in each box are from the same model sample.

yield the lowest single-model perplexity, its prediction on word "York" is still benefited from the other two via averaging.

## 5.2 Image Caption Generation

We next consider the problem of image caption generation, which is a conditional RNN model, where image features are extracted by residual network (He et al., 2016), and then fed into the RNN to generate the caption. We present results on two benchmark datasets, Flickr8k (Hodosh et al., 2013) and Flickr30k (Young et al., 2014). These datasets contain 8,000 and 31,000 images, respectively. Each image is annotated with 5 sentences. A single-layer LSTM is employed with the number of hidden units set to 512.

The widely used BLEU (Papineni et al., 2002),

METEOR (Banerjee and Lavie, 2005), ROUGE-L (Lin, 2004), and CIDEr-D (Vedantam et al., 2015) metrics are used to evaluate the performance. All the metrics are computed by using the code released by the COCO evaluation server (Chen et al., 2015b).

Table 4 presents results for pSGLD/RMSprop with or without dropout. Consistent with the results in the basic language modeling, pSGLD yields improved performance compared to RMSprop. For example, pSGLD provides 2.77 BLEU-4 score improvement over RMSprop on the Flickr8k dataset. By comparing pSGLD with RMSprop with dropout, we conclude that pSGLD exhibits better regularization ability than dropout on these two datasets.

Apart from modeling weight uncertainty, different samples from our algorithm may capture different aspects of the input image. An example with two images is shown in Fig. 4, where 2 randomly chosen model samples are considered for each image. For each model sample, the top 3 generated captions are presented. We use the beam search approach (Vinyals et al., 2015) to generate captions, with a beam of size 5. In Fig. 4, the two samples for the first image mainly differ in the color and activity of the dog, *e.g.*, "tan" or "yellow", "playing" or "running", whereas for the second image, the two samples reflect different understanding of the image content.

## 5.3 Sentence Classification

We study the task of sentence classification on 5 datasets as in (Kiros et al., 2015): *TREC* (Li and Roth, 2002), *MR* (Pang and Lee, 2005), *SUBJ* (Pang and Lee, 2004), *CR* (Hu and Liu, 2004) and *MPQA* (Wiebe et al., 2005). A single-layer bidirectional LSTM is employed with the

Table 5: Sentence classification errors on five benchmark datasets.

| Methods | MR | CR | SUBJ | MPQA | TREC |
|---|---|---|---|---|---|
| RMSprop | $21.26_{\pm1.45}$ | $22.70_{\pm2.20}$ | $8.13_{\pm1.19}$ | $10.60_{\pm1.28}$ | $8.58_{\pm0.79}$ |
| RMSprop + Dropout | $20.33_{\pm0.67}$ | $20.70_{\pm2.22}$ | $7.24_{\pm0.86}$ | $10.66_{\pm0.74}$ | $7.82_{\pm0.66}$ |
| RMSprop + Gal's Dropout | $19.66_{\pm0.60}$ | $20.21_{\pm2.34}$ | $7.52_{\pm1.17}$ | $10.59_{\pm1.12}$ | $8.32_{\pm0.52}$ |
| pSGLD | $20.36_{\pm1.09}$ | $21.18_{\pm1.90}$ | $7.43_{\pm1.21}$ | $10.54_{\pm0.99}$ | $7.88_{\pm0.72}$ |
| pSGLD + Dropout | $\mathbf{19.48}_{\pm0.95}$ | $\mathbf{19.74}_{\pm2.03}$ | $\mathbf{6.61}_{\pm1.06}$ | $\mathbf{10.22}_{\pm0.89}$ | $\mathbf{7.32}_{\pm0.66}$ |

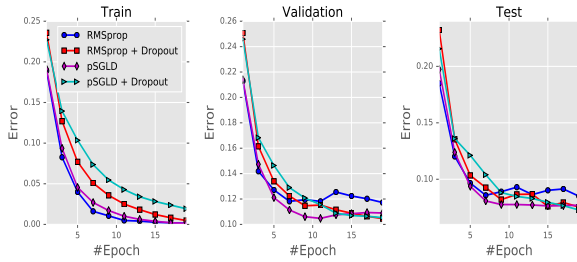

Figure 5: Learning curves on TREC dataset.

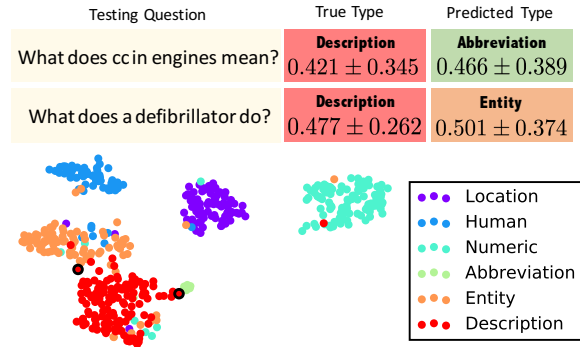

Figure 6: Visualization. Top two rows show selected ambiguous sentences, which correspond to the points with black circles in tSNE visualization of the testing dataset.

number of hidden units set to 400. Table 5 shows the testing classification errors. 10-fold cross-validation is used for evaluation on the first 4 datasets, while TREC has a pre-defined training/test split, and we run each algorithm 10 times on TREC. In addition to (naive) dropout, we further compare pSGLD with the *Gal's dropout*, recently proposed in (Gal and Ghahramani, 2016b), which is shown to be applicable to recurrent layers. The combination of pSGLD and dropout consistently provides the lowest errors.

In the following, we focus on the analysis of TREC. Each sentence of TREC is a question, and the goal is to decide which topic type the question is most related to: *location*, *human*, *numeric*, *abbreviation*, *entity* or *description*. Fig. 5 plots the learning curves of different algorithms on the training, validation and testing sets of the TREC dataset. pSGLD and dropout have similar behavior: they explore the parameter space during learning, and thus coverge slower than RSMprop on the training dataset. However, the learned uncertainty alleviates overfitting and results in lower errors on the validation and testing datasets.

To further study the Bayesian nature of the proposed approach, in Fig. 6 we choose two testing sentences with high uncertainty (*i.e.*, standard derivation in prediction) from the TREC dataset. Interestingly, after embedding to 2d-space with tSNE (Van der Maaten and Hinton, 2008), the two sentences correspond to points lying on the boundary of different classes. We use 20 model sam-

ples to estimate the prediction mean and standard derivation on the true type and predicted type. The classifier yields higher probability on the wrong types, associated with higher standard derivations. One can leverage the uncertainty information to make decisions: either manually make a human judgement when uncertainty is high, or automatically choose the one with lower standard derivations when both types exhibits similar prediction means. A more rigorous usage of the uncertainty information is left as future work.

# 6 Conclusion

We propose a scalable Bayesian learning framework using SG-MCMC, to model weight uncertainty in recurrent neural networks. The learning framework is tested on several tasks, including language models, image caption generation and sentence classification. Our algorithm outperforms stochastic optimization algorithms, indicating the importance of learning weight uncertainty in recurrent neural networks. Our algorithm requires little additional computational overhead in training, and multiple times of forward-passing for model averaging in testing. Future works include improving the testing efficiency for the large-scale RNNs, via learning a single neural network that approximates the model averaging result (Korattikara et al., 2015).

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
