# Peer review of "Scalable Bayesian Learning of Recurrent Neural Networks for Language Modeling"

_ACL 2017 — decision unknown_

[Official Review · Reviewer 1 · rating 4 · confidence 4]
soundness 3 · originality 3 · clarity 4 · impact 4 · substance 4 · appropriateness 5 · meaningful comparison 4 · presentation format Oral Presentation

- Strengths:

a) The paper presents a Bayesian learning approach for recurrent neural network
language model. The method outperforms standard SGD with dropout on three
tasks. 
b) The idea of using Bayesian learning with RNNs appears to be novel. 
c) The computationally efficient Bayesian algorithm for RNN would be of
interest to the NLP community for various applications.

- Weaknesses:

Primary concern is about evaluation:

Sec 5.1: The paper reports the performance of difference types of architectures
(LSTM/GRU/vanilla RNN) on character LM task while comparing the learning
algorithms on the Penn Treebank task. Furthermore, RMSprop and pSGLD are
compared for the character LM while SGD +/- dropout is compared with SGLD +/-
dropout on word language model task. This is inconsistent!  I would suggest
reporting both these dimensions (i.e. architectures and the exact same learning
algorithms) on both character and word LM tasks. It would be useful to know if
the results from the proposed Bayesian learning approaches are portable across
both these tasks and data sets.

L529: The paper states that 'the performance gain mainly comes from adding
gradient noise and model averaging'. This statement is not justified
empirically. To arrive at this conclusion, an A/B experiment with/without
adding gradient noise and/or model averaging needs to be done. 

L724: Gal's dropout is run on the sentence classification task but not on
language model/captions task. Since Gal's dropout is not specific to sentence
classification,  I would suggest reporting the performance of this method on
all three tasks. This would allow the readers to fully assess the utility of
the proposed algorithms relative to all existing dropout approaches.

L544: Is there any sort order for the samples? (\theta_1, ..., \theta_K)? e.g.
are samples with higher posterior probabilities likely to be at higher indices?
Why not report the result of randomly selecting K out of S samples, as an
additional alternative?

Regular RNN LMs are known to be expensive to train and evaluate. It would be
very useful to compare the training/evaluation times for the proposed Bayesian
learning algorithms with SGD+ dropout. That would allow the readers to
trade-off improvements versus increase in training/run times.

Clarifications:
L346: What does \theta_s refer to? Is this a MAP estimate of parameters based
on only the sample s?
L453-454: Clarify what \theta means in the context of dropout/dropconnect. 

Typos:
L211: output
L738: RMSProp

[Official Review · Reviewer 2 · rating 4 · confidence 4]
soundness 3 · originality 3 · clarity 4 · impact 4 · substance 4 · appropriateness 4 · meaningful comparison 4 · presentation format Poster

- Strengths:
1) The paper is trying to bridge the gap between Stochastic Gradient MCMC and
Stochastic Optimization in deep learning context. Given dropout/dropConnect and
variational inference are commonly used to reduce the overfit, the more
systematic way to introduce/analyse such bayesian learning based algorithms
would benefit deep learning community.
2) For language modeling tasks, the proposed SG-MCMC optimizer + dropout
outperforms RMSProp + dropout, which clearly shows that uncertainty modeling
would help reducing the over-fitting, hence improving accuracy.
3) The paper has provided the details about the model/experiment setups so the
results should be easily reproduced.

- Weaknesses:
1) The paper does not dig into the theory profs and show the convergence
properties of the proposed algorithm.
2) The paper only shows the comparison between SG-MCMC vs RMSProp and did not
conduct other comparison. It should explain more about the relation between
pSGLD vs RMSProp other than just mentioning they are conterparts in two
families.
2) The paper does not talk about the training speed impact with more details.

- General Discussion:

[Official Review · Reviewer 3 · rating 4 · confidence 4]
soundness 3 · originality 3 · clarity 4 · impact 4 · substance 3 · appropriateness 5 · meaningful comparison 4 · presentation format Oral Presentation

- Strengths: This paper explores a relatively under-explored area of practical
application of ideas behind Bayesian neural nets in NLP tasks. With a Bayesian
treatment of the parameters of RNNs, it is possible to incorporate benefits of
model averaging during inference. Further, their gradient
based sampling approximation to the posterior estimation leads to a procedure
which is easy to implement and is potentially much cheaper than other
well-known techniques for model averaging like ensembling.  
The effectiveness of this approach is shown on three different tasks --
language modeling, image captioning and sentence classification; and
performance gains are observed over the baseline of single model optimization.

- Weaknesses: Exact experimental setup is unclear. The supplementary material
contains important details about burn-in, number of epochs and samples
collected that should be in the main paper itself. Moreover, details on how the
inference is performed would be helpful. Were the samples that were taken
following HMC for a certain number of epochs after burn in on the training data
fixed for inference (for every \tilda{Y} during test time, same samples were
used according to eqn 5) ? Also, an explicit clarification regarding an
independence assumption that p(D|\theta) = p(Y,X| \theta) = p(Y| \theta,X)p(X),
which lets one use the conditional RNN model (if I understand correctly) for
the potential U(\theta) would be nice for completeness.

In terms of comparison, this paper would also greatly benefit from a
discussion/ experimental comparison with ensembling and distillation methods
("Sequence level knowledge distillation"; Kim and Rush, "Distilling an Ensemble
of Greedy Dependency Parsers into One MST Parser"; Kuncoro et al.) which  are
intimately related by a similar goal of incorporating effects of model
averaging.

Further discussion related to preference of HMC related sampling
methods over other sampling methods or variational approximation would be
helpful.

Finally, equation 8 hints at the potential equivalence between dropout and the
proposed approach and the theoretical justification behind combining SGLD and
dropout (by making the equivalence more concrete) would lead to a better
insight into the effectiveness of the proposed approach.  

- General Discussion: Points addressed above.